# How Urban Expansion Triggers Spatio-Temporal Differentiation of Systemic Risk in Suburban Rural Areas: A Case Study of Tianjin, China

**Jian Tian [1], Suiping Zeng [2,*], Jian Zeng [1] and Sen Wang [1]**

[1] School of Architecture, Tianjin University, Tianjin 300072, China
[2] School of Architecture, Tianjin Chengjian University, Tianjin 300384, China
* Correspondence: spikelet19870706@126.com

**Abstract:** Rapid urban expansion has strongly impacted rural development in China's suburbs. The increasing probability of socio-ecosystem hazards, such as the shrinking and fragmentation of ecological space, the outflow of labor force, the disintegration of traditional society, and the decline in collective economy has become a systemic risk restricting the sustainable development of rural areas in the suburbs. At present, the influence of urban expansion on rural systemic risk in the suburbs is not clear, which is not conducive to putting forward differentiated and targeted strategies for rural revitalization. Therefore, in this study, we propose the ecological, industrial, social, and livelihood elements of rural systemic risk in the suburbs and construct a multi-dimensional risk resistance analysis framework involving functionality, stability, and sustainability. Taking 93 villages in the western suburbs of Tianjin as an example, and using spatial econometric methods such as remote sensing interpretation, GIS analysis, multiple linear regression, and random forest model testing, we analyze the relationship between external transportation construction, urban employment attraction, construction of land growth, rural risk factors, and the dimension of risk resistance. Finally, the influence of urban expansion on the spatial–temporal differentiation of rural systemic risk and the risk management strategy are discussed. The results show that the difference in the urban expansion intensities is the main factor of the spatial differentiation of rural systemic risk in the suburbs. With the acceleration of the land replacement rate between urban and rural areas, the proportion of urban construction of land is increasing, leading to various degrees of change in the rural land use pattern and the ecological security pattern. Meanwhile, because of the urban employment attraction, part of the rural labor force continues to decrease, leading to the spatial differentiation of rural industrial risks and social risks aggravated. Precise risk management strategies are put forward according to the systemic risks in different types of villages. In villages with a high proportion of urban construction land and inefficient land consolidation, ecological restoration projects should be carried out. For villages severely divided by transit roads, internal spatial connections should be strengthened by constructing public transport. For villages with good accessibility, the allocation of rural non-agricultural industries and service facilities should be strengthened to mitigate the impact of urban expansion on the rural social structure. From the perspective of risk management, the research results will provide a basis for making decisions regarding rural public policymaking and spatial resource allocation in the suburbs of developing countries.

**Keywords:** urban expansion; suburbs; systemic risk; risk resistance; differentiation; management policy

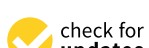

## 1. Introduction

Risk refers to the possibility of an adverse outcome of an event, and systematic risk refers to the multiple risks resulting from the interaction of various interrelated risk factors [1]. The rural systemic risk discussed in this paper refers to the increasing probability of the occurrence of social ecosystem hazards, such as shrinking and fragmentation of

ecological space, labor outflow, disintegration of traditional society, and collective economic recession, under the influence of urban expansion in suburban rural areas.

At present, developing countries are experiencing the stage of rapid urbanization, because of which, suburban villages show high complexity and rapid change and are easy to destroy [2,3]. Rural ecological security, industrial development, social stability, people's livelihood, and other aspects are strongly impacted, resulting in the coexistence of multiple systemic risks [4,5]. To resolve suburban rural systemic risks and realize urban–rural coordination and sustainable development, developing countries must accurately identify suburban rural systemic risks under the impact of urban expansion and interpret their formation mechanism.

The academic circle has carried out research on rural risk in the suburbs, with partial success. Improvements are still needed in three areas. In terms of research content, emphasis is placed on ecological risks and less attention is paid to social, industrial, and livelihood risks. In terms of spatial scale, there is a lack of mesoscale research on the administrative village as a unit. In terms of research methods, there are few studies on the spatial and temporal distribution of risks. Let us look at the three areas in detail.

In terms of research content, ecological, environmental, and health risks are paid the most attention, with warnings regarding environmental pollution, ecological damage, and ecological risk as the research focus. For example, soil and water pollutants sources and prevention measures were proposed through the analysis of the ecological risk index (RI) for soil and water pollution in rural areas caused by urban expansion [6]. ArcGIS spatial analysis technology was used to analyze the risk of rural water damage, and measures were put forward to protect natural vegetation and build rural sewage treatment facilities [7]. Analyzing the relationship between cultivated land use intensity and ecological risk and identifying the change in cultivated land use can help control ecological risk in advance [8]. The above methods help people to understand the changes in rural ecological patterns resulting from urban expansion and lay a foundation for solving the problems of the rural environment and health risks. However, due to the failure to consider the interaction between rural industrial development, social stability, people's livelihood demand, and ecological environment, there is a lack of studies on the impact of various complex factors of urban expansion on rural risk resistance and insufficient analysis of the correlation mechanism of the spatial–temporal differentiation of risk. Therefore, it is difficult to put forward an implementable rural risk management strategy systematically.

In terms of spatial scale, some scholars focus on analyzing large-scale rural temporal and spatial scope, while others analyze rural risks from the perspective of the micro-farmer unit and the livelihood vulnerability of rural communities and residents in the suburbs becomes the focus [9]. For example, on the basis of social interviews, it was suggested that diversified livelihoods should be developed to improve individuals' risk adaptation capacity as natural resources in rural areas decrease day by day [10]. The livelihood vulnerability index (LVI) and factor analysis were used to analyze the livelihood vulnerability of mixed urban and rural communities, and it was pointed out that the rural areas in the suburbs are less dependent on natural resources and the coordinated livelihood model with cities has not yet been formed [11]. The adaptation mechanism of rural social-ecosystemic risks in response to tourism development was put forward after carrying out a household survey of peasants [12]. The above analyses have provided practical help in formulating rural macro-strategies and laid a research foundation for solving the individual problems of farmers. However, there are few studies on the spatio-temporal differentiation of various risks taking villages as data units and there is a lack of mesoscale risk analysis combining the characteristics of different villages. Therefore, it is not easy to formulate public policies adapted to the characteristics of different villages, nor can it provide adequate technical support for the precise allocation of rural public resources.

In terms of research methods, there are standard statistical analysis and index evaluation methods for assessing rural vulnerability. For example, the nonlinear principal component method was used to assess the driving factors of rural socio-economic vulnera-

bility in the suburbs, including employment opportunities, government effectiveness, and natural resources [13]. The abstract statistical classification method was used to analyze the relationship between rural economic diversity and economic risk minimization, and the development of specific types of rural economic activities was proposed [14]. A vulnerability evaluation model was built to analyze the temporal and spatial differentiation characteristics of rural production space vulnerability and summarize the types of rural industrial vulnerability [15]. The above research methods have achieved positive results. However, further research is needed to accurately understand the spatial and temporal distribution characteristics of rural risks and scientifically analyze the influence of urban expansion on various rural risks by combining statistical analysis and spatial measurement methods.

Therefore, in view of the problems in the existing research results, in this paper, we have added the research on social, industrial, and livelihood risks to the research content; strengthened the mesoscale village unit analysis on the spatial scale; and applied the spatial econometric analysis technology in the research method. This study takes villages as data units and uses spatial econometric analysis technology to analyze the spatio–temporal differentiation characteristics and formation mechanism of rural systemic risk in suburban areas under urban expansion. It will lay a theoretical foundation for accurately formulating public policies and allocating public resources to effectively resolve rural systemic risks in the suburbs and provide technical research support.

## 2. Study Context and Data Sources

### 2.1. Research Scope

Tianjin is one of the four municipalities directly under the central government in China and the second-largest city in northern China. From 1988 to 2018, the urbanization rate increased from 48.5% to 83.2%. The land use, social economy, and ecological space of the suburbs of Tianjin changed dramatically [16]. In this study, we selected 93 administrative villages in four towns (Yangliuqing Town, Xinkou Town, Duliu Town, and Yangfengang Town) in the western suburbs of Tianjin as the research area (Figure 1). The research area is located in the hinterland of the Beijing–Tianjin–Hebei city conglomeration, close to the Tianjin–Xiongan New Area spatial development corridor. Compared with other areas in the suburbs of Tianjin, in the western suburbs, urban space expansion is rapid and strongly influences rural development. It is a representative sample for studying the rural systemic risk in suburbs under the impact of rapid urban expansion.

The research scope included the inner suburbs, the middle and far suburbs, and other location types and covered all kinds of functional spaces on the urban fringe. For example, Yangliuqing Town is close to the urban area, so urban and rural space is intertwined and most villages are highly urbanized; the villages in Xinkou Town have developed modern agriculture practices and rural service industries; many villages in Duliu Town are located in the flood detention area and are ecologically sensitive; and the majority of villages in Yangfengang Town are based on traditional agricultural production and the development level is relatively backward. Meanwhile, the number of villages within the research scope (93) conformed to the research needs of rural risk in urban suburbs on the mesoscale. Diversified spatial types and a reasonable number of villages provided essential conditions for studying the spatial differentiation characteristics of rural systemic risk in the suburbs.

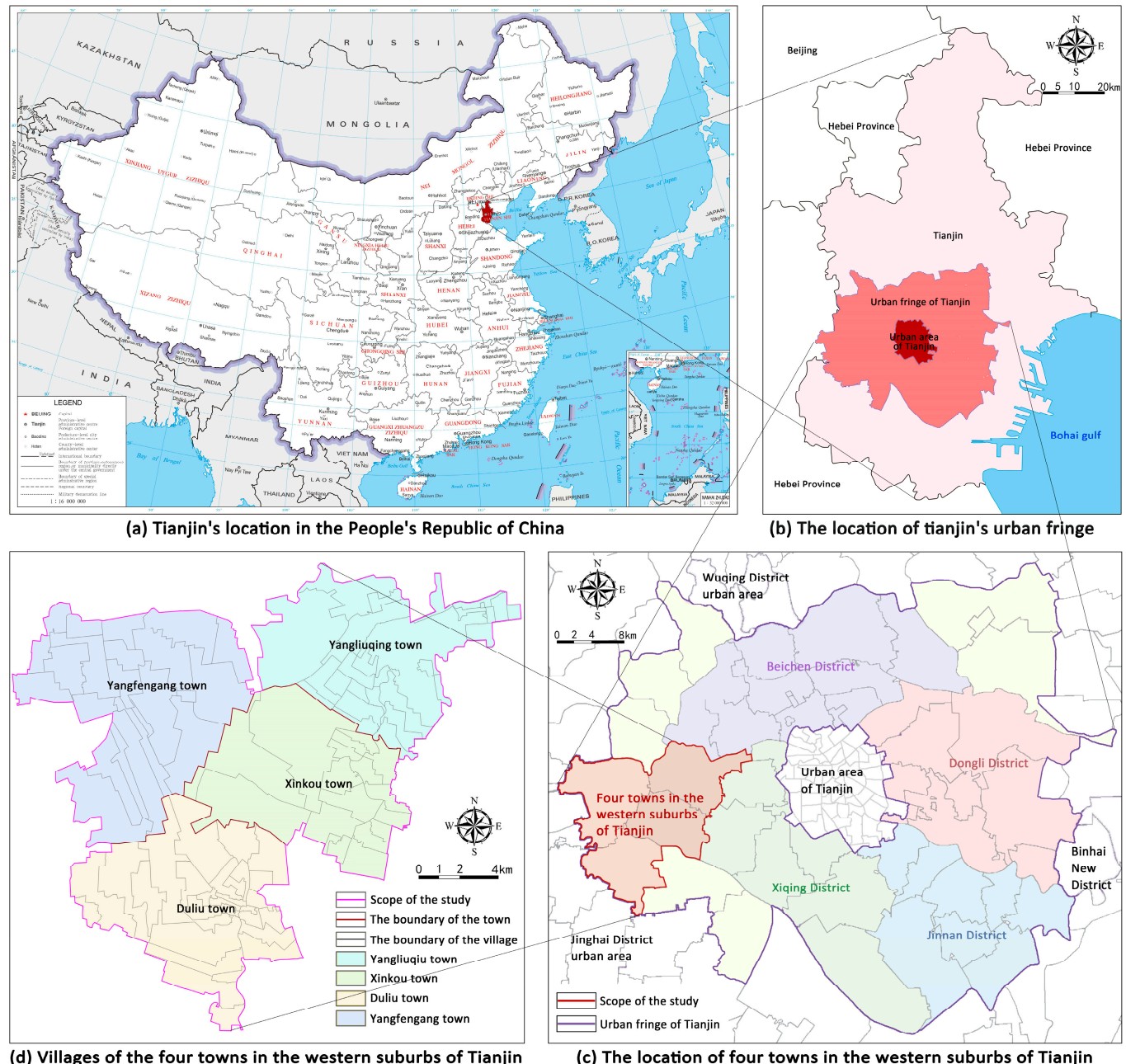

**Figure 1.** Location of the research area. (**a**) The location of Tianjin in China, (**b**) the extent of the suburbs of Tianjin, (**c**) the extent of the four towns in the western suburbs of Tianjin, and (**d**) the administrative boundaries of the 93 villages in the western suburbs of Tianjin (source: self-drawn).

### 2.2. Data Sources

### 2.2.1. Acquisition of Land Use and Road Traffic Data

Using Geospatial Data Cloud, we obtained Landsat 7 and Landsat 8 remote sensing image data (1998–2018, with a spatial resolution of 30 m and 15 m, respectively). We apply ENVI image data processing technology to interpret remote sensing image data as land use spatial vector data that can be processed by the GIS platform. The vector data of all roads in rural areas in the western suburbs of Tianjin were downloaded via the Universal Maps Downloader, which provides data support for calculating traffic accessibility and external road density (Table 1).

**Table 1.** Data information.

| Data | Utility | Data Source |
| --- | --- | --- |
| Land use data | Spatial stability, ecological resource richness, proportion, and change in construction land can be calculated, and rural ecological risks can be analyzed. | Landsat 7 and Landsat 8 remote sensing images, which can be downloaded through a geospatial data cloud platform and interpreted through ENVI |
| Boundary data of administrative villages | It can support the study of spatial differentiation with the village as a unit. | Database of Bazhou City Planning Bureau and Tianjin Urban Planning and Design Institute |
| Road data | It can be used to calculate traffic accessibility, external traffic road density, etc., and analyze the degree of influence of urban expansion. | Omnipotent electronic map download software |
| Socio-economic data | It includes data on population, industry, employment, land transfer, income, collective economy, public service facilities, etc., which can be used to analyze the risks to the rural industry, society, and people's livelihood. | Questionnaire surveys, interview records, and Internet data search (http://www.tcmap.com.cn/tianjin/xiqingqu.html, accessed on 9 November 2019) |

Source: self-created.

### 2.2.2. Acquisition of Socio-Economic Data

This research mainly involved a literature review of books, papers, and Internet materials and adopted social research methods, such as questionnaire surveys and interview records. Questionnaires were sent to the village managers in each administrative village to acquire data related to rural social, industrial, and livelihood factors and analyze the development situation of the village (a questionnaire was sent to each village within the scope of the study, for a total of 93 questionnaires, and 93 valid questionnaires were recovered). Meanwhile, village cadres and some villagers in each village were interviewed to understand the village's collective or private economic development. Then, the reliability of data, such as the labor outflow level, residents' income, the land transfer ratio, and rural service facilities, was verified.

## 3. Methodology

### 3.1. Research Framework

The impact of urban expansion on different villages in the suburbs is spatially and temporally different. Therefore, the systematic risk resistance capacity and various risk factors of rural areas also show different characteristics across villages. The impact of urban expansion on suburban villages is mainly reflected in the impact of transit traffic construction, the attraction of urban employment to the rural labor force, and the embedding of urban construction land into rural areas [17]. Therefore, based on GIS technology, in this study, we first calculate the density of transit roads in each village, the commuting time from each village to the urban area, and the proportion of urban construction land areas embedded in the countryside and identify the spatial characteristics of urban expansion to the suburban countryside. Meanwhile, on the basis of multi-source data statistics and GIS technology, we analyze the spatial differentiation characteristics of industrial, social, livelihood, and ecological factors of rural systemic risk and the spatial differentiation characteristics of functional, stability, and sustainability factors of rural resistance to risk. Then, we analyze the correlation mechanism between urban expansion and spatial differentiation of rural systemic risk. We use Pearson correlation analysis to extract influential correlation factors and use multiple linear regression analysis and a random forest model test to determine the core correlation factors. Finally, we use the logic analysis method to analyze the mechanism of action between core correlation factors and clarify the spatial differentiation law of risk

factors and the ability to resist suburban rural systemic risk under the influence of urban expansion (Figure 2).

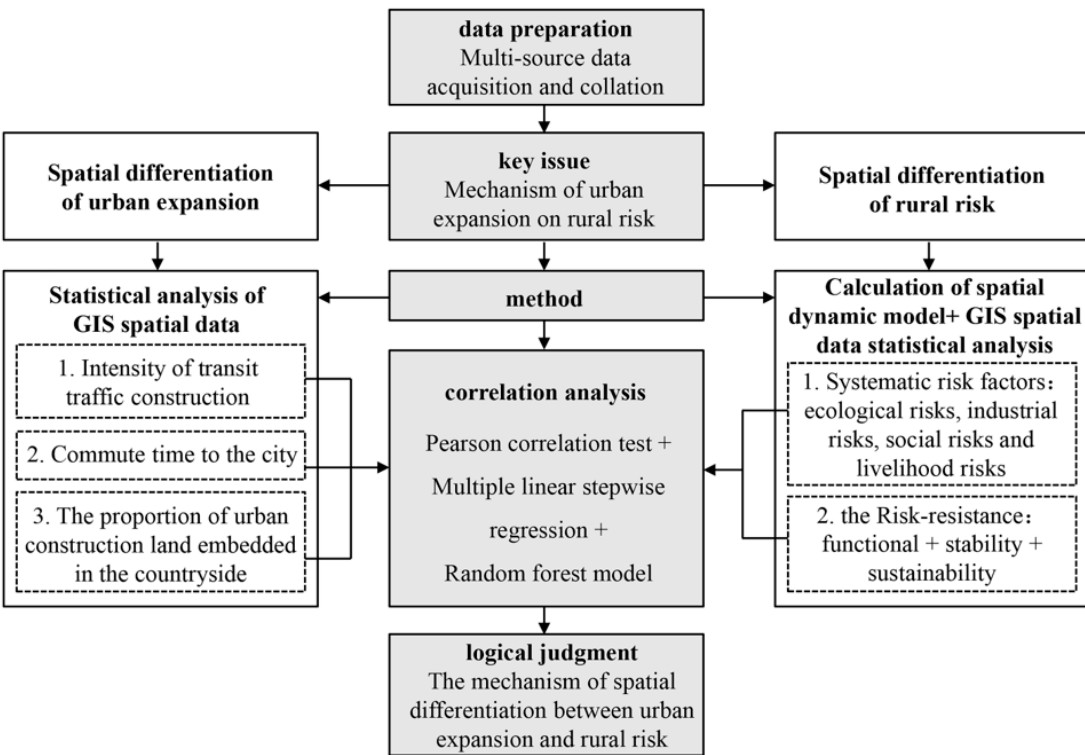

**Figure 2.** Research framework of spatio-temporal differentiation and mechanism of rural systemic risk under the influence of urban expansion (source: self-drawn).

*3.2. Observed Variable*

3.2.1. Observational Variables of the Impact of Urban Expansion on Suburban Rural Risk

The factors of urban expansion that influence rural systemic risk are mainly reflected in the following aspects: ① The construction of transit traffic outside the city will interfere with the ecological pattern and industrial development of the suburbs, so the density of external roads in each village can be selected as the corresponding observation variable of transit traffic construction [18]. The higher the road density, the greater the interference intensity of transit traffic to the village. ② The changes in service radiation and employment attraction intensity in urban areas will have an impact on rural population composition, industrial development, and social stability, which can be represented by the commuting time from the village to the central city [19]. The shorter the commute time, the better the accessibility, indicating that the village is more influenced by the attraction of urban employment and public services. ③ The embedding of urban construction land into rural areas refers to replacing rural land with urban land for industry, universities, and large markets. It tends to reduce the space of rural ecology, living, and agricultural production, which can be represented by the proportion of urban construction land embedded in suburban villages [20]. ④ Urban expansion may cause pollution in the rural environment, and the pollution degree can be represented by the air quality index (AQI) of the atmospheric environment and the related index of water pollution [21]. In this paper, we mainly discuss the first three aspects.

3.2.2. Observational Variables of Rural Systemic Risk Factors in Suburbs

Rural systemic risk influences industry, society, people's livelihood, ecology, and other contents (latent variables). According to the representativeness and accessibility of the related data, we select the observed variables of the following risk factors (Table 2). The

variable "proportion of agricultural employees" can be selected in terms of industrial elements. If the value of this variable is too high, it reflects that the industry type is too single; a too-low value it indicates that the foundation of endogenous-characteristic industries in rural areas is insufficient and these industries are vulnerable to the impact of urban industries [22]. The variable "proportion of labor outflow" can be selected in terms of social factors. A high proportion of labor outflow indicates that the human base of social development is weak and the social structure is unstable [23]. The variable "employment diversity index" can be selected in terms of livelihood factors. A low value of this variable indicates that the employment type is single and employment is unstable [24]. The "average area of ecological patch" index can be selected in terms of ecological factors. A low value of this index indicates that ecological patch fragmentation is severe and ecological risk is high [25].

**Table 2.** Observational variables of systemic risk factors in suburban villages.

| Latent Variable | Observational Variable | Risk Characterization |
| --- | --- | --- |
| The sustainable development level of the rural industry | The proportion of agricultural employees [22] | Too high or too low |
| | The proportion of land transfer [2,17] | Not suitable for farming needs |
| | Arable land per capita [8] | Too little |
| | Main types of non-agricultural industries [12,17] | Low end |
| | Main types of farming [17] | Low economic benefit |
| | The development level of collective and private economy [17] | Low |
| | The per capita area of industrial land [15] | Too large |
| The stable level of the rural social structure | The spillover ratio of labor outflow [23] | High |
| | The magnitude of population change [17,24] | Rapid |
| | The density of the population [13,26] | Low |
| | Housing vacancy rate [12] | High |
| | The frequency of group activities organized [17] | Low |
| | The percentage of support group coverage [17] | Low |
| The level of prosperity and facilities in the countryside | Employment diversity index [24] | A single type of employment |
| | The number of primary and secondary schools within 1 km [17,22] | Insufficient |
| | The number of health facilities within 1 km [17] | Insufficient |
| | The number of bus stops within 1 km [17,22] | Insufficient |
| | Commuting time to and from the city [15,24] | Long |
| | Average annual household income [11,12] | Low |
| Rural ecological pattern stability and resource richness | The average area of the ecological patch [25] | Plaque too fragile |
| | The percentage of ecological vegetation coverage [3,27] | Low |
| | The percentage of water area coverage [3,17] | Low |
| | The percentage of farmland coverage [8] | Low |
| | The rate of overall spatial change [27] | Rapid |
| | Ecological space resource index [17,28] | Less ecological resources |

Source: self-created.

### 3.2.3. Observational Variables of Rural Risk Resilience in Suburban Areas

The functionality, stability, and sustainability of the system development can characterize rural risk resistance. The more functional the rural development, the more developed the economy, the richer the villagers, and the stronger the ability to resist economic risks. The more stable the rural development, the more diverse the employment of residents, the more stable the population and spatial changes, and the stronger the inheritance of local culture, indicating that the rural ability to resist social risks is strong. The higher the sustainability of rural development, the more perfect the natural ecosystem service function, and the stronger the ability of the rural area to resist environmental risks. Therefore, we selected these three aspects as potential variable indicators to measure rural risk resistance ability and the corresponding observation variable indicators for each type of potential variable according to the typicality and accessibility of variables [29–31] (Table 3).

**Table 3.** Observational variables characterizing rural risk resistance.

| The Dimension of Ability to Resist Risk | The Subdivision Dimension of Risk Resistance | Resilience Targets against Risk | Primary Observed Variable |
|---|---|---|---|
| System functionality | The abundance of the villagers' lives | Good economic performance of peasant households | Annual household income [11,12] |
| | The level of collective economy | Prosperous collective industries | The annual income of village collective [17,29] |
| | Job diversity | Strong livelihood security | Employment diversity index [24] |
| System stability | The effectiveness of ensuring people's well-being | Guaranteed public services | The number of facilities per kilometer radius [17,30] |
| | The reasonableness of population composition | Stable social structure | The level of labor outflow [23] |
| | The stability of land use | Stable spatial structure | The rate of spatial change [27,31] |
| System sustainability | The effectiveness of ecosystem services | Good function of the ecosystem | Ecological space resource index [17,28] |
| | The integrity of the ecological security pattern | Secure ecological patterns | The average area of the ecological patch [25] |

Source: self-created.

### 3.3. Methods of Spatial Econometric Analysis

The research on the correlation between urban expansion and spatial differentiation of suburban rural systemic risk involves analyzing the correlation between the projected values of certain urban expansion elements in each rural space unit and the numerical matrices of risk elements in each space unit (Figure 3). It mainly involves spatial econometric analysis methods, such as spatial information statistics, spatial change rate model calculation, and spatial data analysis.

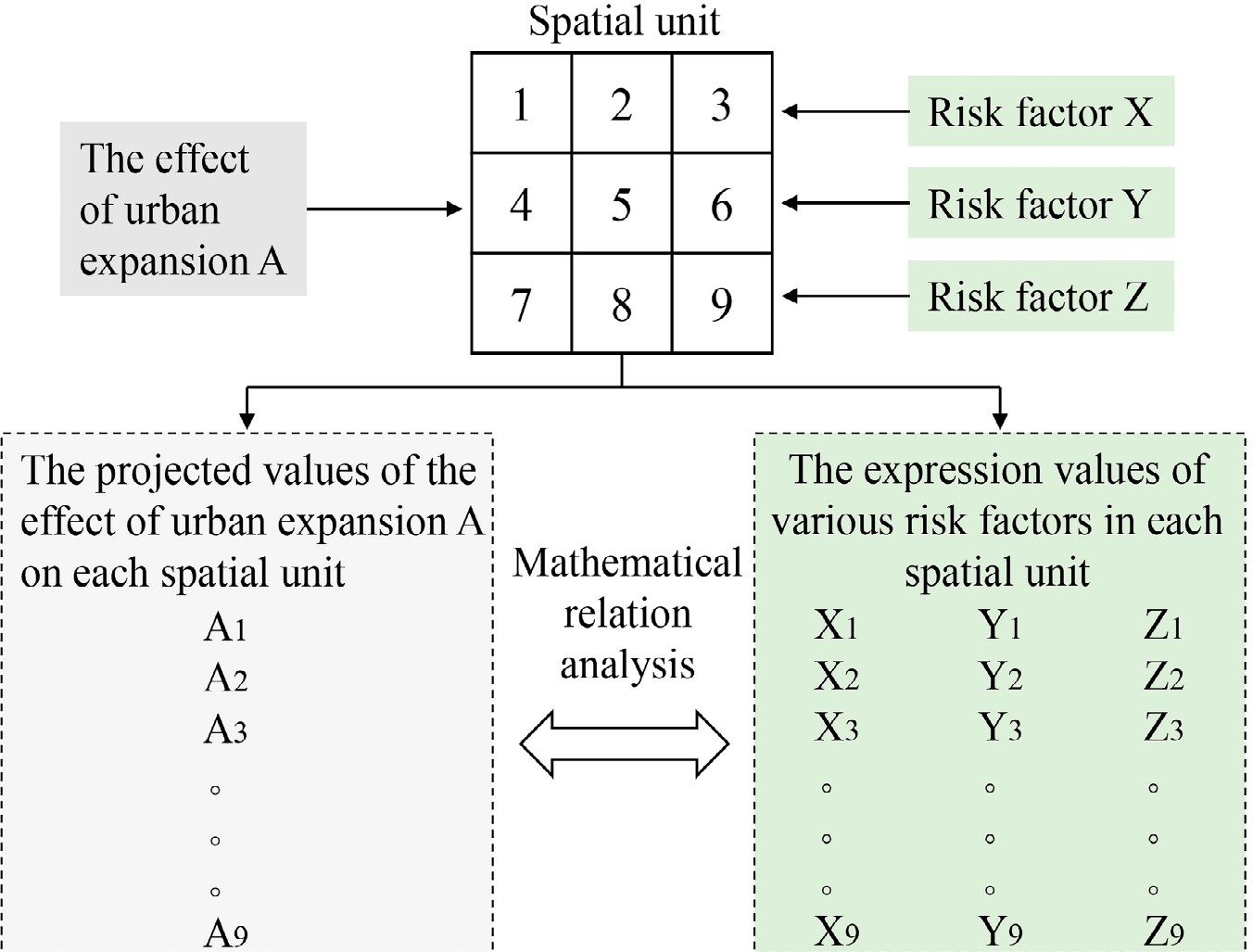

**Figure 3.** The principle of mathematical correlation analysis between urban expansion and the rural systemic risk (source: self-drawn).

3.3.1. Method of Calculating Spatial Information Statistics and the Spatial Change Rate Model

The spatial and temporal distribution characteristics of the rural systemic risk can be identified by processing land use data interpreted by remote sensing and calculating social and economic data through the GIS platform. The spatial change rate model is used to quantitatively analyze the rates of change of various types of land use in suburban rural areas and analyze the ecological risk pattern from the perspective of stability. It includes the rates of change of single land use and comprehensive land use. The rate of change of single land is expressed in Formula (1).

$$K = \frac{U_a + U_b}{U_1} \times \frac{1}{t_2 - t_1} \times 100\% \tag{1}$$

Here, $K$ is the rate of change of single land use, $t_2 - t_1$ is the duration of the study period, $U_1$ is the initial annual area of a certain type of land within the research scope, $U_a$ is the increased area (absolute value) of this type of land during this study period, $U_b$ is the area (absolute value) reduced by this type of land in this study period, and $U_a + U_b$ is the total dynamic change in the area of this kind of land use in this research period.

The comprehensive land use change rate refers to the overall rate of change in the scales of all types of land use in the study area in a certain period, expressed in Formula (2).

The change rate of extensive land use reflects the overall stability of land use: the higher the dynamic attitude, the more unstable the space and the higher the ecological risk.

$$LC = \frac{\sum_{i=1}^{n} \Delta LA_{(ij)}}{\sum_{i=1}^{n} LA_{(ij)}} \times \frac{1}{t_2 - t_1} \times 100\% \qquad (2)$$

Here, $LC$ represents the rate of change of comprehensive land use, $\Delta LA_{(ij)}$ represents the absolute value of the land area (category $i$ land and non-category $i$ land) during the study period, $LA_{(i,t_1)}$ is the area of the category $i$ land in the study area at the initial monitoring time $t_1$, and $t_2 - t_1$ represents the duration of the research period [27].

### 3.3.2. Correlation Analysis Method of Risk Factors Based on the Multiple Regression Model

Multiple linear stepwise regression methods can be used to screen out the optimal set of variables under the condition of mutual influence of multiple variables in the system [32]. Therefore, this method is suitable for screening rural risk factors strongly associated with a specific urban expansion function in this study as a basis for further logical analysis. The expression of the multiple linear regression model is as follows (Formula (3)):

$$A = \beta_0 + \beta_1 D_1 + \beta_2 D_2 + \beta_3 D_3 + \cdots + \beta_P D_P + \varepsilon \qquad (3)$$

where $A$ is the target variable (the observed variable of a specific urban expansion effect in this study); $\beta_0$ is the regression constant; $\beta_1, \beta_2 \ldots \beta_P$ is the regression coefficient (which can represent the strength of the correlation in this study); $D_1, D_2 \ldots D_P$ is the influential observational variable of various risk factors in the affected villages; and $\varepsilon$ is a random error [33].

### 3.3.3. Method of Risk Factor Correlation Analysis Based on the Random Forest Model

Random forest is a kind of advanced algorithm based on a decision tree. Its important function is to identify the importance of complex system elements [34]. This fits the analysis demand of correlation intensity between urban expansion and various elements of rural systemic risk in this study. In this study, the out-of-pocket data method of random forest is used to rank the correlation intensity between a particular urban expansion effect and various risk factors in the suburbs and countryside to screen out the core elements associated with the urban expansion effect logical analysis.

## 4. Results

### 4.1. Spatial and Temporal Differentiation of Urban Expansion to the Suburban Countryside

Our study shows that during the period of rapid urbanization (1988–2018), the expansion effect of Tianjin city to the western suburbs was prominent and the growth of urban construction land, the density of transit roads, transportation accessibility, and other spatial factors changed significantly. The hot spots of spatial change shifted from the inner suburbs to the outer suburbs and were mainly concentrated in the villages around the towns or along the traffic corridors.

### 4.1.1. Changes in the Density of Suburban Transit Roads

From 1988 to 2018, the external transport construction in the western suburbs of Tianjin showed apparent spatial heterogeneity among different villages. In terms of railways, an east–west line was added (Tianjin–Baoding High-speed Railway). Three new expressways emerged, among which the Beijing–Shanghai expressway runs through the study area from north to south. Meanwhile, the road density in the connecting direction between each town and the central urban area improved significantly. The road construction sequence ran from the inner suburbs (eastern) to the outer suburbs (western). The density of transit roads generally presented the spatial distribution characteristics of "dense inner suburbs, sparse outer suburbs" and "high intensity around towns" (Figure 4).

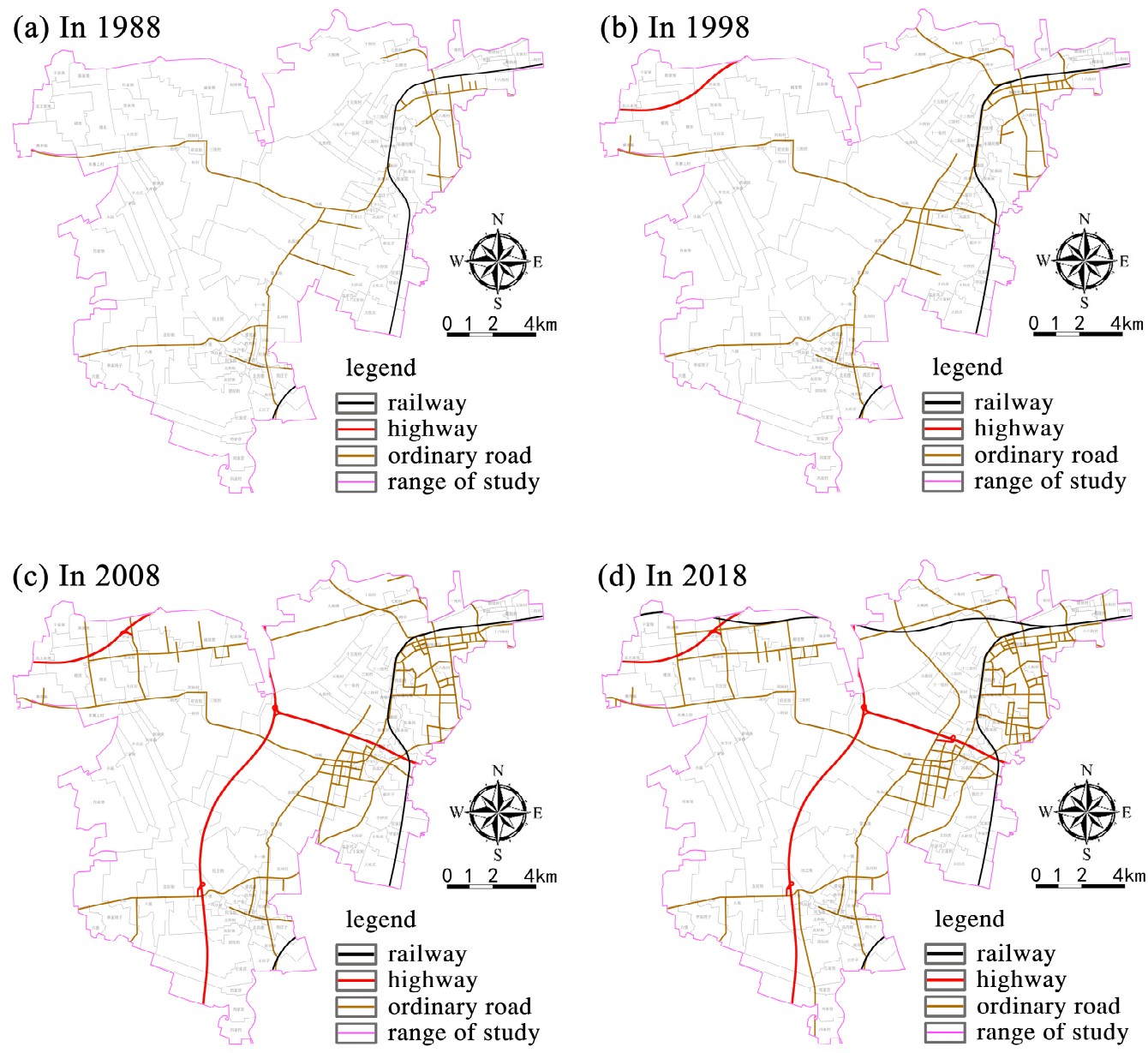

**Figure 4.** Changes in the rural external traffic network in the western suburbs of Tianjin during various periods of urban expansion (source: self-drawn).

4.1.2. Changes in Rural–Urban Accessibility

The accessibility from rural areas to urban areas can reflect the attractiveness of urban employment services to rural areas. From 1988 to 2018, the transport accessibility of the western suburbs of Tianjin continued to change and showed significant spatial differences. The range of high accessibility (less than 30 min commuting time to the urban area) continued to expand from several villages in the northeast near the urban center to most of the areas covered by the study. The rural accessibility to urban areas in the southwest, which are far from the traffic corridor, was relatively low. Figure 5 shows the changes in accessibility. The color changes from darker to lighter represent the spatial difference in accessibility from weak to strong.

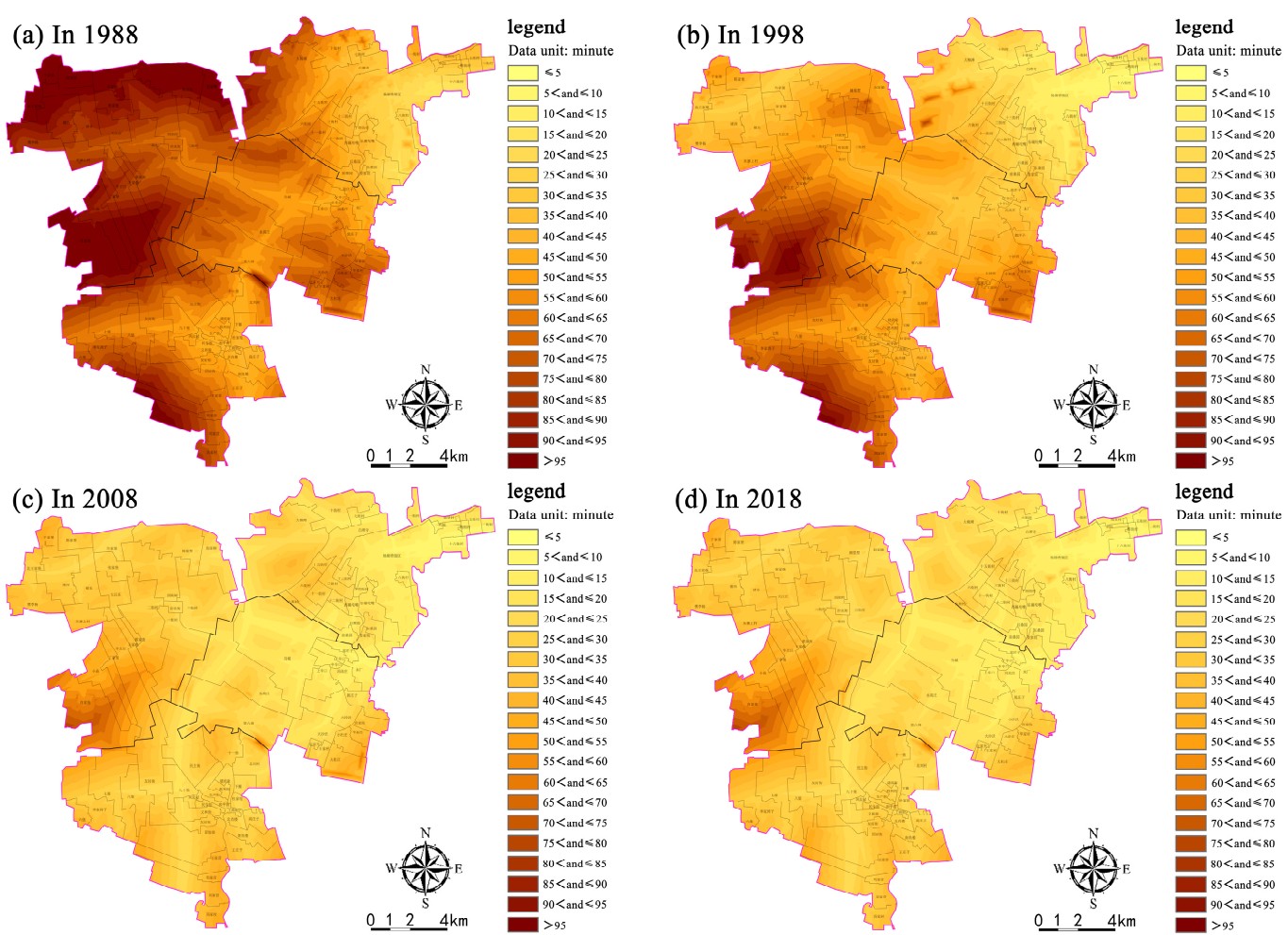

**Figure 5.** Change in rural accessibility to urban areas in the western suburbs of Tianjin during various periods of urban expansion (source: self-drawn).

4.1.3. Temporal and Spatial Changes in Urban Land Embedding in the Suburban Countryside

From 1988 to 2018, the embedding of urban construction land into rural areas in the western suburbs of Tianjin kept expanding and presented firm spatial heterogeneity. In 1988, the urban construction land in the western suburbs of Tianjin was concentrated in Yangliuqing Town, which was close to the urban area. Subsequently, urban construction land continued to expand from inner suburbs to outer suburbs. By 2018, urban construction land dominated the construction land of villages in Yangliuqing Town and a large amount of urban construction land also appeared in villages in other towns. The hotspots of urban construction land change were relatively concentrated, developing from point and line to strip and block and mainly distributed in the inner suburbs of the city (east of Yangliuqing Town) and industrial parks of each town (north of Yangfengang Town and south of Xinkou Town) (Figure 6).

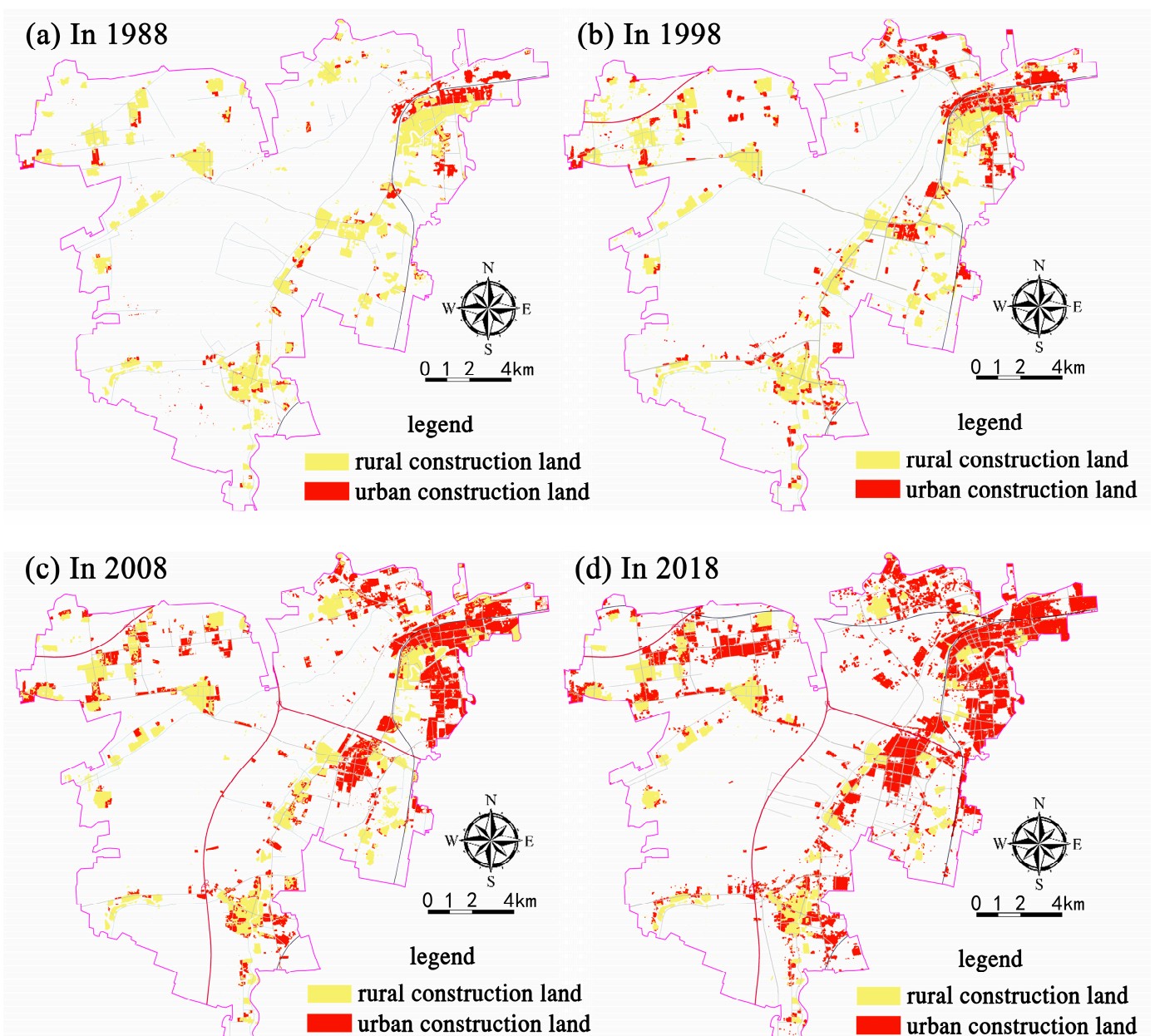

**Figure 6.** Changes in urban and rural construction land in the western suburbs of Tianjin during various periods of urban expansion (source: self-drawn).

*4.2. Spatial Differentiation of Rural Systemic Risk under the Impact of Urban Expansion*

The results show that under the influence of urban expansion, various systemic risk factors in rural areas show different degrees of spatial differentiation and the risk resistance capacity of different villages is significantly different.

4.2.1. Spatial Differentiation Characteristics of Systematic Risk Factors in the Suburban Countryside

Industry, society, people's livelihood, and the ecology of rural systemic risk in the western suburbs of Tianjin showed different degrees of spatial differentiation. Regarding industrial risk factors, land transfer in some middle and outer suburbs lagged behind the demand for agricultural development, which is not conducive to the large-scale and efficient development of agriculture. The collective economic development of some villages in the southwest was low, which is not conducive to the sustainable development of rural industries. In terms of social risks, the proportion of migrant workers around some towns

and villages in the northwest was high, resulting in a significant outflow of the labor force, which is not conducive to the stable development of population and society. The collective governance capacity of the villages around some towns and villages in the inner suburbs was poor, and the cohesion and collective action capacity of the villages were insufficient. Regarding livelihood risks, some villages in central and western areas lacked facilities such as education and medical care, making it challenging to provide the villagers security in terms of health and education. At the level of ecological risk, the ecological patches in the villages in the inner suburbs, along the traffic corridors, and around towns were seriously fragmented and the ecological space resources supporting the sustainable development of rural areas had been destroyed. On the basis of the above risk factors, the villages with a high level of systemic risk were concentrated in two types of areas. One was the surrounding villages of Yangliuqing Town, which is close to the central urban area. The second was the outlying villages, represented by the western part of Yangfengang Port (Figure 7).

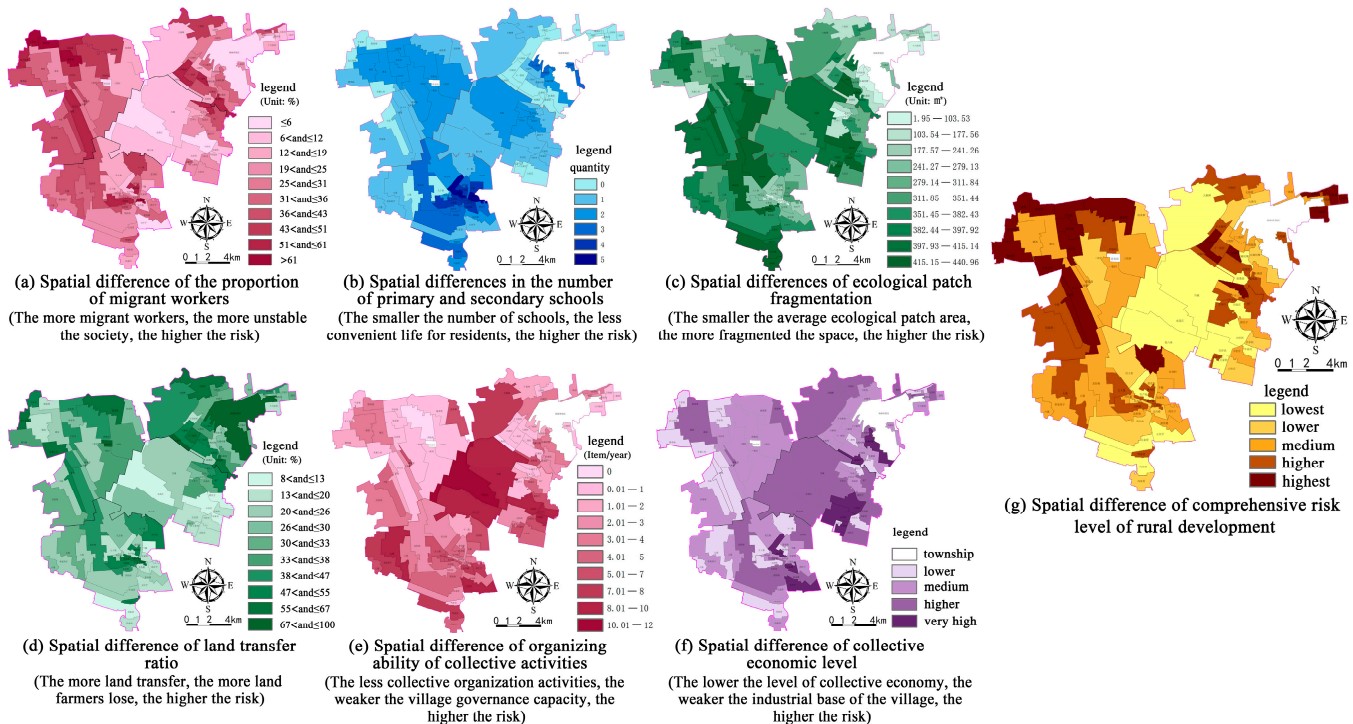

**Figure 7.** Spatial differentiation of rural systemic risk factors in the western suburbs of Tianjin (source: self-drawn).

### 4.2.2. Spatial Differentiation of Rural Risk Resistance in the Suburban Countryside

The rural areas in the western suburbs of Tianjin showed distinct spatial differences in different dimensions of risk resistance. Regarding function, the per capita income of some villages in the northwest was low, the residents were not wealthy, and the economic development performance was not high. In terms of stability, the rate of spatial change of villages in the inner suburbs and around towns was faster and the pattern of rural ecological security and industrial development was unstable. Some villages in the east and southwest had low employment diversity, and the single employment structure made it challenging to withstand sudden market risks. In terms of sustainability, the proportion of ecological land in the inner suburbs, industrial areas, and villages around towns was low. The resources of ecosystem services on which rural sustainable development depends were less, which is not conducive to the sustainable development of rural endogenous power. Overall, the villages with low risk resistance were concentrated in two areas. One is the surrounding villages of Yangliuqing Town, which is close to the central city. The second is the outlying villages, represented by the western part of Yangfengang (Figure 8).

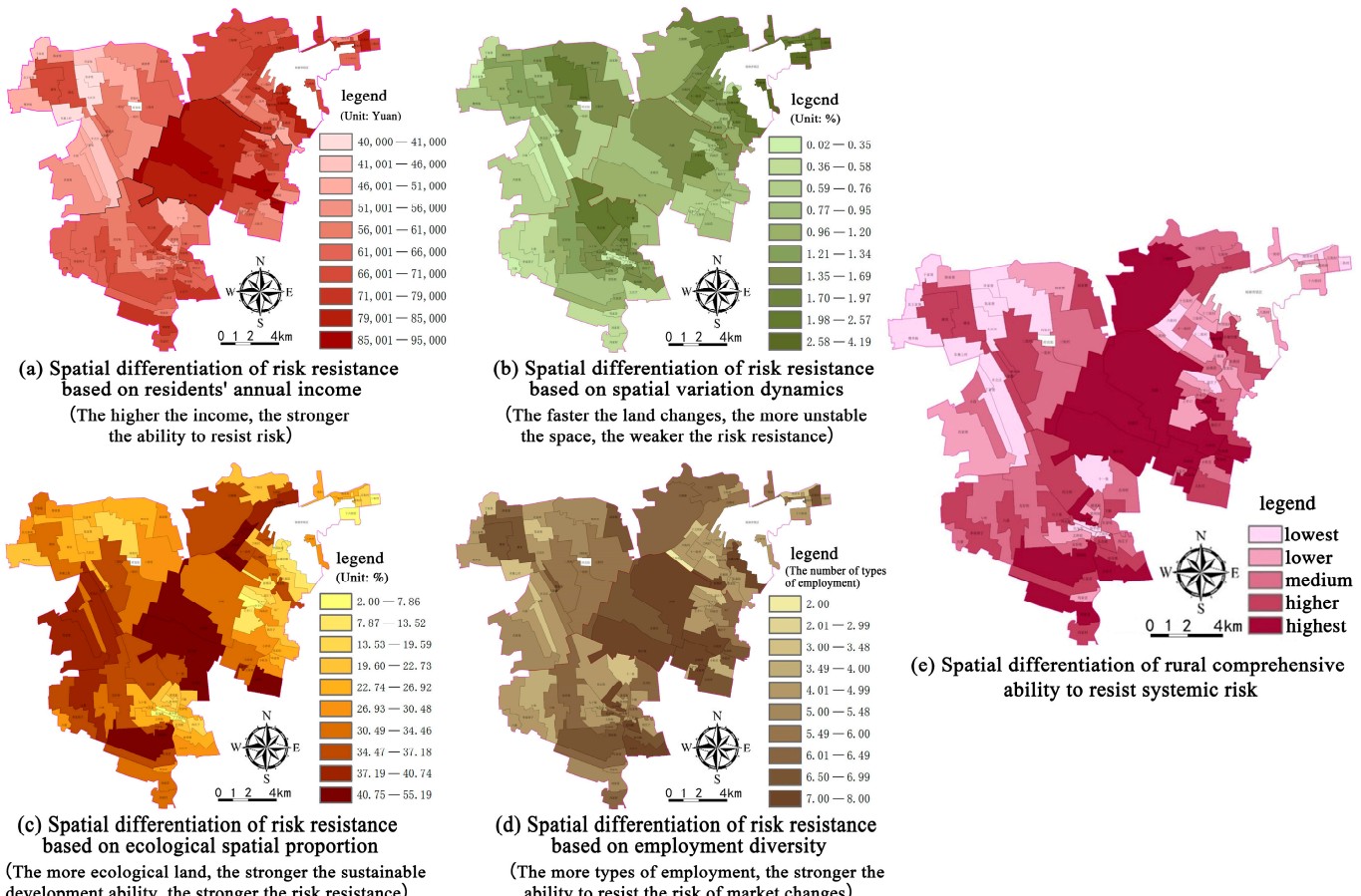

**Figure 8.** Spatial differentiation of rural resilience to risk in the western suburbs of Tianjin (source: self-drawn).

### 4.3. Correlation and Mechanism between Urban Expansion and Rural Risk Resistance

By analyzing the correlation between the urban expansion effect (transit road construction, urban employment service attraction, urban construction land embedding into rural areas), and various dimensions of rural risk resistance, we comprehensively analyzed the mechanism of urban expansion effect on suburban rural risk resistance.

#### 4.3.1. Correlation between the Density of Transit Roads and Rural Risk Resistance

Studies show that transit road construction (the observed variable is road density) is significantly correlated with the ecological sustainability dimension of rural risk resistance (the observed variable is ecological space resource index) (Figure 9; Pearson correlation coefficient is over 0.6, and the reasonable degree of composite linear equation is within the acceptable range). The denser the transit road, the more serious the disturbance to the ecosystem, and the worse the rural ecological ability to resist risks. This shows that transit traffic disturbance is one of the main factors influencing rural ecological risk resistance and has a weak correlation with the economic function and social stability of the rural area.

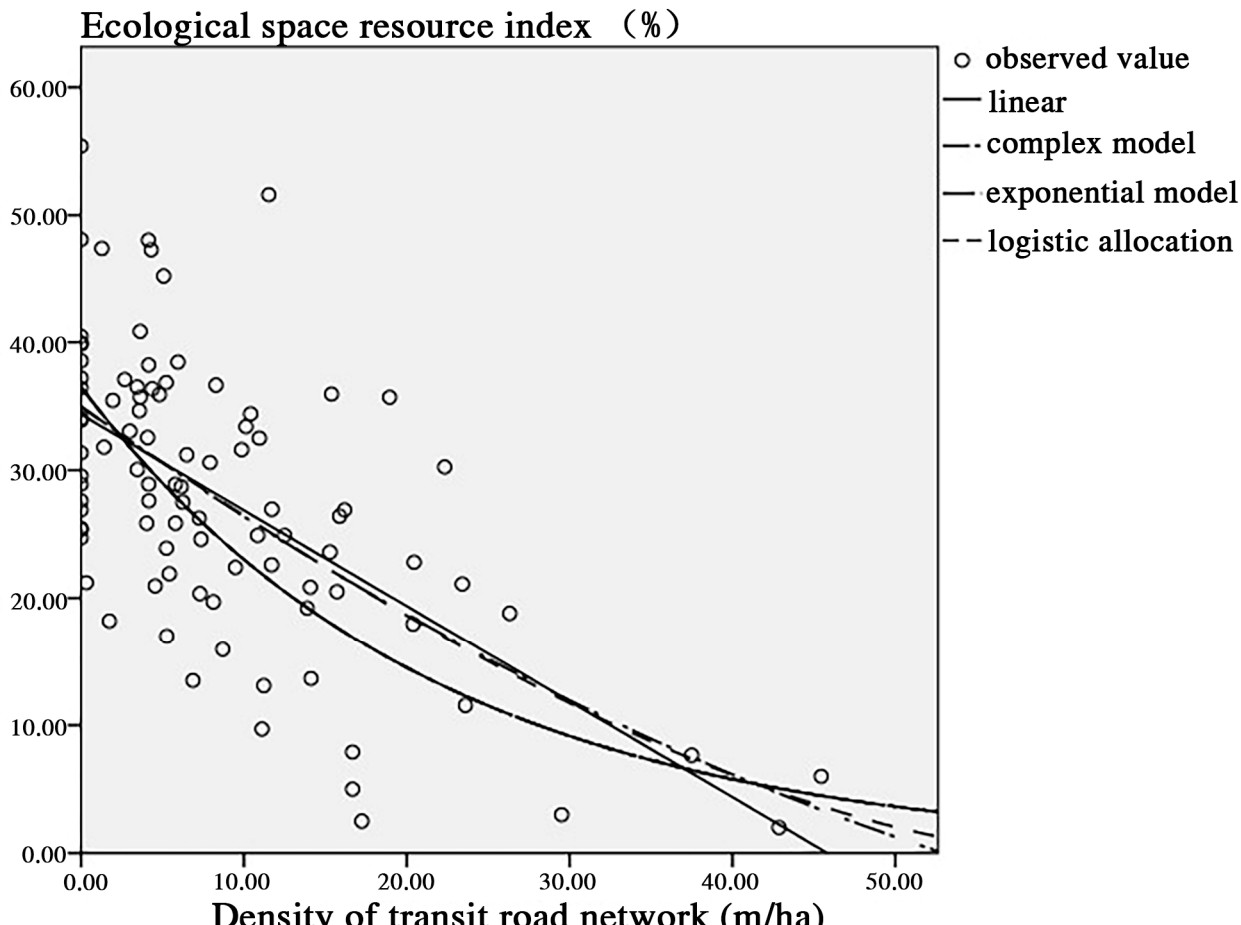

(a)

**correlation**

|  |  | Density of transit road network (m/ha) | Ecological space resource index (%) |
|---|---|---|---|
| **Density of transit road network (m/ha)** | **Pearson correlation** | 1 | −0.619[**] |
|  | **Significance (two tails)** |  | 0.000 |
|  | **N** | 93 | 93 |
| **Ecological space resource index (%)** | **Pearson correlation** | −0.619[**] | 1 |
|  | **Significance (two tails)** | 0.000 |  |
|  | **N** | 93 | 93 |

**\*\*. The correlation was significant at 0.01 level (two-tailed)**

(b)

**Figure 9.** (**a**) Equation fitting analysis of the transit road density and the ecological space resource index (source: self-drawn). (**b**) Correlation analysis between the transit road density and the ecological space resource index (source: self-drawn).

### 4.3.2. The Correlation between Urban Employment Attraction and Rural Risk Resistance

The study shows a positive correlation between the difference in accessibility to urban areas and the level of income of villagers, which indicates that the convenience of obtaining urban employment has a particular impact on the risk resistance of the suburban rural economy. Meanwhile, the accessibility difference has a significant negative correlation with

the sustainability of the rural ecological environment and social stability. The higher the accessibility, the faster the spatial change, the lower the proportion of ecological space, and the lower the ecological stability. The more convenient the urban employment choice, the more the rural labor outflow, indicating that the attractiveness of urban employment services is one of the main factors affecting the stability of the rural social structure.

### 4.3.3. Correlation between Urban Construction Land Embeddedness and Rural Risk Resistance

Studies show that the proportion of urban construction land in rural areas has a significant negative correlation with the sustainability of the rural ecology (Figure 10), while the correlation with the risk resistance of the functional dimension of the rural economy is weak. The higher the proportion of urban construction land embedded in rural areas, the faster the spatial change, and the lower the proportion of ecological space, the lower the ecological stability and sustainability. This indicates that the embedding of urban construction land is one of the main factors affecting the suburban rural risk resistance resulting from ecological sustainability.

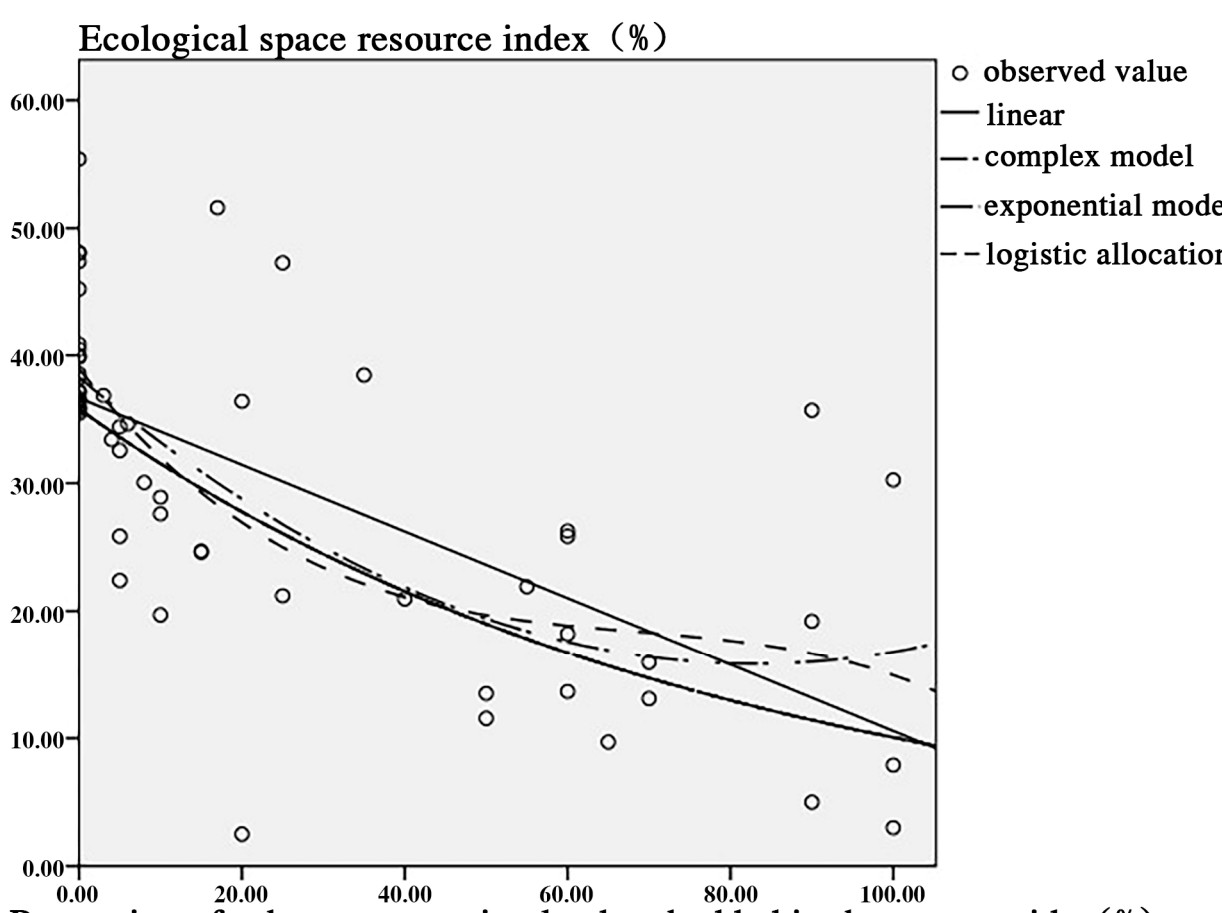

(a)

**Figure 10.** *Cont.*

**correlation**

|  |  | Proportion of urban land embedded in countryside (%) | Ecological space resource index (%) |
|---|---|---|---|
| Proportion of urban land embedded in countryside (%) | Pearson correlation | 1 | −0.672** |
|  | Significance (two tails) |  | 0.000 |
|  | N | 56 | 56 |
| Ecological space resource index (%) | Pearson correlation | −0.672** | 1 |
|  | Significance (two tails) | 0.000 |  |
|  | N | 56 | 93 |

**.The correlation was significant at 0.01 level (two-tailed)

(**b**)

**Figure 10.** (**a**) Equation fitting analysis of the proportion of urban construction land and the ecological space resource index (source: self-drawn). (**b**) Correlation analysis between the proportion of urban construction land and the ecological space resource index (source: self-drawn).

*4.4. Screening and Mechanism of the Core Factors of Urban Expansion and Rural Systemic Risk*
4.4.1. Screening and Mechanism of Correlation Factors between Transit Road Density and Rural Systemic Risk

Firstly, SPSS Pearson correlation analysis was used to test the data correlation and the factors influencing the density of transit roads that were screened from various elements of rural systemic risk. Then, on the basis of multiple regression and the random forest model, the core correlation factors between road density and rural risk factors and their mechanism were further analyzed.

(1) Screening and mechanism analysis of influential correlation factors

The results show that the density of transit roads is negatively correlated with the average area of the ecological patch, the proportion of farmland, and the proportion of ecological vegetation, which shows that the density of transit roads aggravates the ecological risk by dividing the ecological space. Meanwhile, transit road density is positively correlated with population density and various public service facilities, indicating that this factor promotes the agglomeration of population and public service facilities by influencing transportation accessibility and changing the distribution pattern of social and livelihood risks. There is a significant negative correlation between the density of transit roads and the proportion of agricultural employees, which shows that areas with a high road density have a relatively high level of non-agricultural industry development and a relatively low proportion of agricultural employees, which reduces the risks to industry and livelihood from the perspective of diversified industry development (Table 4).

As per the above analysis, the transit road density is significantly correlated with variables of rural risk factors such as ecological patch fragmentation, land use, a permanent population, agricultural employment, village accessibility, and public service facilities. Therefore, the above variable types were extracted as influential correlation factors of the rural systemic risk under the influence of transit traffic construction. The mechanism is as follows: the higher the density and intensity of transit roads, the higher the proportion of construction land, and the lower the proportion of farmland and ecological vegetation reflects the interaction between transit roads and construction land expansion. Roads often pass through residential and industrial areas with more travel demands, and road construction improves spatial accessibility and promotes more population and construction activities. Moreover, the proportion of ecological space decreases compared with that of other villages, increasing ecological risks.

**Table 4.** Statistical analysis of the correlation between transit traffic network density and rural risk factors.

| Variables of Rural Risk Factors | Pearson Correlation | Significance (Two Tails) |
|---|---|---|
| The proportion of agricultural employees | −0.363 | 0.000 |
| The proportion of land transfer | 0.202 | 0.052 |
| Arable land per capita | −0.139 | 0.184 |
| Main types of non-agricultural industries | —— | —— |
| Main types of farming | —— | —— |
| The development level of collective and private economy | —— | —— |
| The per capita area of industrial land | 0.001 | 0.995 |
| The spillover ratio of labor outflow | 0.032 | 0.760 |
| The magnitude of population change | 0.202 | 0.053 |
| The density of the population | 0.689 | 0.000 |
| Housing vacancy rate | −0.146 | 0.162 |
| Employment diversity index | −0.062 | 0.553 |
| The number of primary and secondary schools within 1 km | 0.359 | 0.000 |
| The number of health facilities within 1 km | 0.219 | 0.035 |
| Commuting time to and from the city | −0.432 | 0.000 |
| Average annual household income | 0.071 | 0.497 |
| The average area of the ecological patch | −0.649 | 0.000 |
| The percentage of ecological vegetation coverage | −0.249 | 0.016 |
| The percentage of water area coverage | −0.189 | 0.069 |
| The percentage of farmland coverage | −0.635 | 0.000 |
| The rate of overall spatial change | 0.137 | 0.189 |
| Ecological space resource index | −0.619 | 0.000 |

Source: self-created

(2)    Screening and mechanism analysis of core correlation factors

On the basis of the effective correlation factor types of rural systemic risk under the influence of transit traffic construction, the stepwise regression method of multiple linear regression analysis was adopted to screen out the core factors (the factors removed in the stepwise regression process can be regarded as non-core factors that have significant correlation and are indirectly affected by core factors).

According to the model abstract in the analysis results (the table on the left in Figure 11), the R-value of the regression equation is 0.792 and the adjusted $R^2$ value is 0.611, indicating that the effectiveness of the interpretation adaptability of the regression results reaches 61.1%, which is within a reasonable range. According to the variance analysis (the table on the right in Figure 11), the F-value test value is 37.071 (Sig. = 0.000), indicating a high significance, proving that the regression equation has good effectiveness. After extracting regression constants and regression coefficients (Table 5), the multiple regression equation is as follows: $A = 6.412 + 0.399 S_2 + 0.280 S_8 + 0.165 L_2 - 0.237 L_5$.

| | Abstract of model | | | |
|---|---|---|---|---|
| model | R | R square | Adjusted for R squared | Standard skew error |
| 1 | 0.689[a] | 0.474 | 0.469 | 6.72862 |
| 2 | 0.760[b] | 0.578 | 0.568 | 6.06573 |
| 3 | 0.778[c] | 0.606 | 0.593 | 5.89037 |
| 4 | 0.792[d] | 0.628 | 0.611 | 5.75948 |

a. forecast: (constant), the population density (person/ha)
b. forecast: (constant), the population density (person/ha), proportion of construction land (%).
c. forecast: (constant), the population density (person/ha), proportion of construction land (%) and commute time to downtown (min)
d. forecast:(constant), the population density (person/ha), proportion of construction land (%), commute time to downtown (min), the number of schools within a kilometer

Anova analysis [a]

| model | | Sum of squares | df | Mean square | F | significant |
|---|---|---|---|---|---|---|
| 1 | regression | 3717.955 | 1 | 3717.955 | 82.121 | 0.000 |
| | residual | 4119.968 | 91 | 45.274 | | |
| | total | 7837.923 | 92 | | | |
| 2 | regression | 4526.548 | 2 | 2263.274 | 61.514 | 0.000 |
| | residual | 3311.375 | 90 | 36.793 | | |
| | total | 7837.923 | 92 | | | |
| 3 | regression | 4749.941 | 3 | 1583.314 | 45.633 | 0.000 |
| | residual | 3087.982 | 89 | 34.696 | | |
| | total | 7837.923 | 92 | | | |
| 4 | regression | 4918.825 | 4 | 1229.706 | 37.071 | 0.000 |
| | residual | 2919.098 | 88 | 33.172 | | |
| | total | 7837.923 | 92 | | | |

a. dependent variable: density of transit road network (m/ha)

**Figure 11.** Multiple linear regression analysis between transit road density and practical factors of rural systemic risk (source: self-drawn).

**Table 5.** The correlation coefficient between road density and risk factors based on multiple linear regression.

| Model | Non-Standardized Coefficient | | Normalization Coefficient | T | Sig. |
|---|---|---|---|---|---|
| | B | Standard Error | Beta | | |
| (Constant) | 6.412 | 2.420 | | 2.650 | 0.010 |
| The density of the population ($S_2$) | 0.345 | 0.076 | 0.399 | 4.544 | 0.000 |
| The proportion of construction land ($S_8$) | 0.101 | 0.034 | 0.280 | 2.986 | 0.004 |
| Commuting time to and from the city ($L_5$) | −0.241 | 0.078 | −0.237 | −3.088 | 0.003 |
| The number of primary and secondary schools within 1 km ($L_2$) | 1.166 | 0.517 | 0.165 | 2.256 | 0.027 |

Source: self-created.

Meanwhile, the random forest model was used to calculate the correlation weight coefficients of each influential correlation factor relative to the density of transit roads (Table 6). The results show that the density of transit roads correlates with traffic accessibility, the proportion of construction land, ecological patch fragmentation, the proportion of farmland, population density, and other risk factors.

**Table 6.** Correlation weight statistics of each influential factor correlated with road density based on the random forest model.

| Effective Correlation Factor | Connection Weights | Sequence |
|---|---|---|
| The average area of the ecological patch | 10.55 | 3 |
| The proportion of construction land | 13.02 | 2 |
| The percentage of farmland coverage | 9.96 | 4 |
| The percentage of ecological vegetation coverage | 1.52 | 8 |
| The density of the population | 8.72 | 5 |
| The proportion of agricultural employees | 2.47 | 7 |
| Commuting time to and from the city | 20.58 | 1 |
| The number of primary and secondary schools within 1 km | 2.96 | 6 |
| The number of health facilities within 1 km | 0.92 | 9 |

Source: self-created.

On the basis of the analysis results of multiple linear regression and the random forest model, we determined the core correlation factors of the transit road density and analyzed the mechanism of transit traffic construction on the core risk factors. Open national, provincial, and county roads in transit transport significantly improve the accessibility of villages along the route, promoting population and the agglomeration of public facilities, helping to reduce livelihood risks and defuse industrial risks. Meanwhile, transit traffic seriously

reduces rural production, living, and ecological space; aggravates the fragmentation of ecological patches; reduces ecological space resources; and intensifies ecological risks.

### 4.4.2. Correlation and Mechanism of Urban Employment Service Attraction and Rural Systemic Risk

Using the same process as that for the above analysis, we used SPSS Pearson correlation analysis to select the factors influencing the attractiveness of urban employment services (commuting time to urban areas) from various elements of rural systemic risk. The results show that the commuting time from a village to an urban area is significantly correlated with rural systemic risk factors, such as ecological patch fragmentation, land use structure, population change, housing vacancy, per capita cultivated land, agricultural employment, construction land change, and residents' income. Therefore, the above variable types were extracted as influential correlation factors between urban employment service attraction and rural systemic risk factors.

Multiple regression and random forest model analysis was used to explore the mechanism of the core factors of rural systemic risk related to the attractiveness of urban employment services. The results show that per capita cultivated land, per capita residential area, the proportion of construction land, and the speed of spatial change have a high correlation with the attractiveness of urban employment services. The mechanism of action is as follows: the villages with higher accessibility, made attractive by part-time employment opportunities and high-quality service facilities, have a more extensive range of population change (reduced social stability), a more vigorous demand for construction land growth, and a faster change of construction land (low ecological stability), increasing social and ecological risks. Meanwhile, with the displacement of arable land and the increase in urban population, the per capita arable land and the per the capita residential area in villages with high accessibility decline, leading to the disappearance of endogenous industries and the entry of urban non-agricultural industries, bringing great uncertainty to the development of rural industries.

### 4.4.3. Correlation and Mechanism between Urban Construction Land Embedding and Rural Systemic Risk Factors

Using the same analysis as the above, we screened the influential factors related to embedding urban construction land into rural areas from various elements of rural systemic risk through SPSS Pearson correlation analysis. The study shows that the proportion of urban construction land in villages is significantly correlated with ecological patch fragmentation, land use, agricultural employment, and per capita industrial land area. Therefore, the above variable types are extracted as factors influencing rural systemic risk under urban construction land embedding.

Using multiple regression and random forest model analysis, we explored the core factors of rural systemic risk related to urban construction land embedding. The results show that the embedment of urban construction land correlates with per capita industrial land, ecological patch fragmentation, the proportion of farmland, and agricultural employment. The mechanism is as follows: because of the large scale of urban construction land, rural land use has changed significantly. The increase in industrial land leads to a decrease in the proportion of farmland, the intensification of ecological patch fragmentation, an increase in the spatial change rate, and a significant increase in ecological risk. Meanwhile, the embedding of urban construction land has changed the rural industrial structure and residents' employment structure, reduced the original rural agricultural production space, and led to a decline in the proportion of agriculture. In addition, with the change in the implanted external functions, the types of non-agricultural industries will also change constantly, leading to an increase in rural industrial development and related risks.

## 5. Discussion

### 5.1. The Influence Mechanism of Urban Expansion on the Risk Resistance of Suburban Countryside

The effect of urban expansion can change suburban rural risk resistance and promote the spatial differentiation of risk resistance among villages. A glance at a summary of the effects of urban expansion on the spatial differentiation of rural risk resistance (Table 7) shows that urban expansion has no significant effect on the economic function dimension of rural risk resistance but has a strong impact on the stability and ecological sustainability dimension.

**Table 7.** Statistical analysis of the impact of urban expansion on spatial differentiation of rural resilience to risk.

| Dimensions of Resilience to Risk / Urban Expansion | Functional Dimension | | | Stability Dimension | | | Sustainability Dimension | |
|---|---|---|---|---|---|---|---|---|
| | The Affluence of the Villagers | The Prosperity of Collective Industries | Diverse Employment Options | Ensuring People's Well-Being | Reasonable Labor Composition | Stable Land Use | Ecological Service Guarantee | Ecological Security Pattern |
| The density of transit roads | Non-significant | Non-significant | Non-significant | Non-significant | Negative significance | Highly negative significance | Highly negative significance | Highly negative significance |
| The attraction of urban employment and services | Positive significance | Positive significance | Highly positive significance | Non-significant | Highly negative significance | Highly negative significance | Negative significance | Negative significance |
| The embedding of urban construction land | Non-significant | Non-significant | Negative significance | Non-significant | Negative significance | Highly negative significance | Highly negative significance | Highly negative significance |

Source: self-created.

In general, the negative impact of urban expansion on the suburban rural risk resistance is dominant. The density of suburban transit roads and the embedding of urban construction land into rural space leads to a sharp drop in the ecological space resources supporting the development of rural characteristics; the pattern of ecological security is destroyed, the ecological stability and sustainability of rural risk resistance decline, and the ecological risk is prominent. The positive impact of urban expansion on rural risk resistance is that improving the spatial accessibility of suburbs leads to more industrial development and creates more employment opportunities for some villages with high accessibility, improves the income of some villagers, and improves the risk resistance from the dimension of economic function.

### 5.2. Influence Mechanism of Urban Expansion on the Suburban Rural Systemic Risk Factors

Urban expansion has a series of direct and indirect impacts on suburban rural systemic risk factors. The main ones are as follows: (1) The intensity of transit traffic construction directly changes the proportion of rural construction land and the rate of spatial change, and then affects the spatial differentiation of the degree of ecological space fragmentation and ecological space resources. (2) The continuous strengthening of employment service attraction in urban areas directly increases the speed of spatial change and the proportion of construction land and increases the risk of ecological space being reduced and fragmented. Meanwhile, improved industrial development and employment opportunities directly influence the differentiation of livelihood risk factors such as cultivated land and resident income, affecting the differentiation of social risk factors such as population change and housing vacancy and bringing the risk of an unstable labor structure in the rural areas. (3) The embedding of urban construction land directly increases the rate of rural spatial change, the degree of ecological space fragmentation, and ecological risks and then promotes a change in industrial risk factors such as cultivated land resources, industry, and agricultural employment (Figure 12).

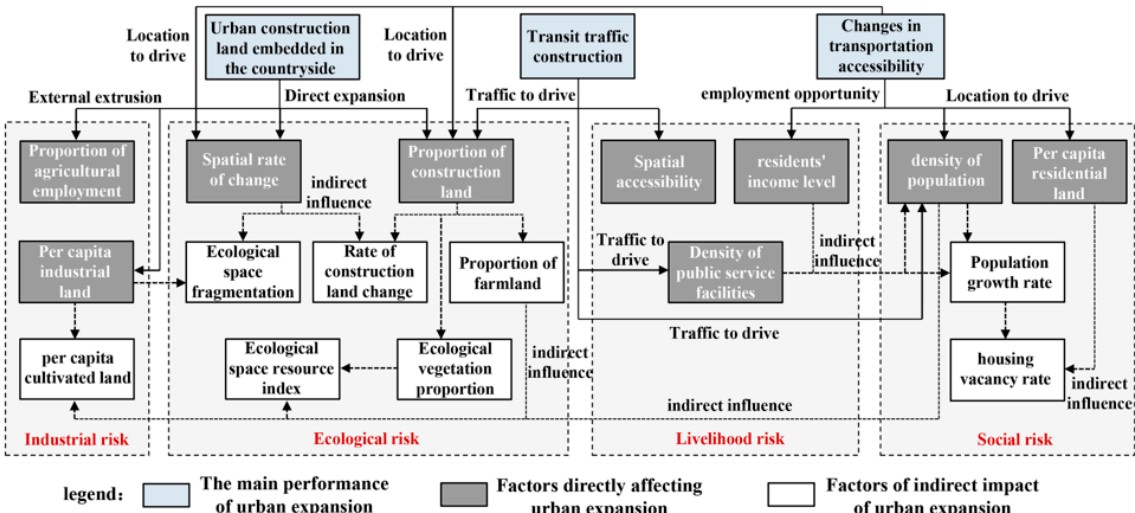

**Figure 12.** The influence mechanism of urban expansion on the suburban rural systemic risk factors (source: self-drawn).

Urban expansion has the maximum influence on ecological risks in suburban areas, influencing the maximum number of factors and having intensive action paths. Urban expansion mainly promotes the spatial differentiation of ecological risk factors by changing the rate of land use change and the proportion of construction land. Furthermore, along with the spatial differentiation of transportation accessibility, means of production, population and employment, public service facilities, and other factors, the spatial differentiation of industry, society, and people's livelihood between villages is deepening.

## 6. Conclusions

In this study, we focused on the suburban rural systemic risk under rapid urbanization in developing countries. Taking the rural areas in the western suburbs of Tianjin, China, as an example, we used multi-source data technology and spatial econometric analysis methods such as the spatial change rate model, multiple linear regression equation, and the random forest model to identify the spatial difference created by urban expansion to suburban villages. Spatial differentiation of suburban rural systemic risk elements and risk resistance was studied, and the correlation between urban expansion and spatial differentiation of rural systemic risk and its mechanism were analyzed.

The results show that the difference in urban expansion intensity is the main factor leading to the spatial differentiation of systematic risk in suburban villages. The negative impact of urban expansion on suburban rural risk resistance is dominant. The construction of suburban transportation and urban construction land embedded in rural areas has reduced ecological space resources supporting the development of rural characteristics, destroying the ecological security pattern, and reducing the stability and ecological sustainability of rural risk resistance. Among all kinds of factors of rural systemic risk, urban expansion has the most concentrated influence on ecological risk factors of suburban villages. The spatial differentiation of ecological risk factors can be promoted mainly by changing the rate of land use change and the proportion of construction land. Meanwhile, part of the rural labor force continues to be lost to urban employment, leading to the spatial differentiation of rural industrial risks and social risks increasingly aggravated.

Differentiated risk management strategies were put forward according to the effect of urban expansion and the risk characteristics of each village. For example, for villages with a high proportion of urban construction land and inefficient land consolidation, ecological restoration projects should be carried out and urban construction land should be strictly controlled in order to be implemented into villages through ecological space use control. For villages with a high density of transit roads, the spatial segmentation effect caused

by transit traffic facilities should be alleviated by constructing internal roads and public transport. Meanwhile, for villages with good accessibility to urban areas and prominent concurrent employment, rural non-agricultural employment and service facilities should be precisely allocated to mitigate the impact of urban expansion on rural human resources and social structure. The above-differentiated risk management strategies of suburban villages are compatible with the spatial heterogeneity of urban expansion effect and rural risks. These strategies can provide an essential decision-making basis for formulating scientific public policy and spatial resource allocation schemes in suburban rural areas.

**Author Contributions:** J.T.: Conceptualization, Methodology, Data collection, Data curation, Writing— original draft preparation and editing. S.Z.: Visualization, Writing—review & editing, Funding acquisition. J.Z.: Supervision, Investigation, Writing—review & edit. S.W.: Writing—review & edit. All authors have read and agreed to the published version of the manuscript.

**Funding:** This study was supported by Tianjin Philosophy and Social Science Planning Project "Study on risk identification and resilience enhancement strategy of Tianjin rural industry under double cycle pattern". Grant number: TJGL21-013.

**Conflicts of Interest:** The authors declare no conflict of interest.

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
