# Peer review of "How Urban Expansion Triggers Spatio-Temporal Differentiation of Systemic Risk in Suburban Rural Areas: A Case Study of Tianjin, China"

_land, doi:10.3390/land11111877_

Round 1
Reviewer 1 Report
Please find attached review comments on introduction part, figures and the conclusions/discussion section.
There are many sentences with unclear expressions and meanings. The text needs to be rewritten.

Reviewer 2 Report
I have reviewed the manuscript titled "How does urban expansion trigger Spatio-temporal differentiation of systemic risk in suburban rural areas: A case study of Tianjin, China".
The basic science of this paper is conducted in a good way and is of an appropriate standard. The author and his team write this paper according to the journal's scope and modern trends. I am glad to review this paper because paper is very interesting according to my research interest area. In the article " How does urban expansion trigger Spatio-temporal differentiation of systemic risk in suburban rural areas: A case study of Tianjin, China", the authors apply spatial econometric analysis methods such as remote sensing interpretation, GIS analysis, multiple linear regression, and random forest model tests are applied to investigate mechanism of urban expansion on the spatial and temporal differentiation of rural systemic risk. This manuscript utilized long-term Landsat images to map urban growth and land-cover changes in Tianjin city of China. The writing of the manuscript looks good overall. I see the novelty in both scientific findings and methodological approaches. The authors clearly state the scientific significance of mapping urban growth and land cover changes in these areas. The manuscript shows a clear picture of the scientific foundation, structure, and focus and clarity of argumentation. The objectives are very clearly outlined in the introduction, the used datasets are well described with sufficient detail.
Minor revision.
The paper is very well organized but, I have a few concerns/suggestions, particularly centring on the English language needs some improvement.
More recent referencing should be added and increase the number of citations.
Line 557: The author wrongly mentioned the heading Discussion instead of Conclusion.
The Conclusion needs some revisions and should be more precise and shorter.
It would be great if future work/recommendations is added to this paper to help and guide city planning and similar research.
Reviewer 3 Report
How does urban expansion trigger spatio-temporal differentiation of systemic risk in suburban rural areas: A case study of Tianjin, China
Dear Authors
This communication paper is conducted in a good way and is of appropriate standard. The author and his team write this paper according to journal scope. I reviewed this paper and I found there are some minor and major mistakes. The structure and the language of the manuscript is in a good shape but there are some long sentences and passive voice sentences. The author need to check language of the manuscript during revision. I am going to suggest major revision at this level based on the revision. I hope the author will modify and highlight the significance and resubmit again in this journal.
I am agree with title of the manuscript.
Line 12-14: Need to revise these lines.
Split long sentences
Check introduction section, there is no sematic connection between paragraphs.
Line 100-103: Split long sentence and rewrite these lines.
Split study area and data sources
Explain the sub figures in caption of figure 1.
Line 136: Check the numbering of heading. It should be 2.2.1: Acquisition of land use and road traffic data.
Line 137: Add space between Landsat 7 and Landsat 8
Line 138: accuracy of 30m and 15m: What do you mean accuracy. Is this accuracy or spatial resolution?
Line 139: Why you added Envi Image definition here. If you proceed the satellite data in ENVI. You should explain here. Otherwise no need to add irrelevant material in the manuscript.
Check footnotes policy of journal. If there is no policy then add reference at proper place.
What type of Landsat data were you used in this research. Add in the table 1.
In figure 5: I can’t see the legend of these sub figures.
Check all figures
The quality of the figures are not according to journal criteria.
Why author write discussion two times?
Merge both discussion section.
In revised version. I ll follow just my own comments.
Best Regards
Round 2
Reviewer 1 Report
Now everything is fixed, thank you
Reviewer 3 Report
Accept in current form